# STAR: Summary Token-guided Attention and Routing for Efficient Long-Context Reasoning

## Abstract

The quadratic complexity of standard attention poses a significant bottleneck for Large Language Models (LLMs) in processing long sequences and integrating information across distant contexts. To overcome this limitation, we propose Summary Token-guided Attention and Routing (STAR), a novel and efficient attention mechanism that selectively retrieves and refines relevant context through a three-stage, coarse-to-fine process. In the Intra-Chunk Abstraction stage, a special summary token appended to each chunk captures local semantics via attention. In the Inter-Chunk Routing stage, a query attends to all summary tokens to identify the most relevant chunks. Finally, the Token-Level Refinement stage applies fine-grained attention over the original tokens within those chunks to enrich contextual representation. Compared to global dense attention, STAR significantly reduces computational cost as input length grows, while preserving the model's ability to reason over long-range dependencies. Experiments on challenging long-context benchmarks show that STAR consistently outperforms existing approaches to enhance long-text processing capabilities of LLMs.

## 1 Introduction

Large Language Models (LLMs) have become a foundational technology in modern natural language processing (NLP), driving significant progress across a wide range of tasks, including document summarization (Zhang et al., 2024b), code generation (Chen et al., 2021), and question answering (Arefeen et al., 2024). Recent models such as GPT-4 (OpenAI, 2023) and LLaMA (Touvron et al., 2023) exemplify this progress, demonstrating impressive language understanding and generation capabilities. These advances have fueled the development of powerful real-world applications in diverse domains, including financial document analysis (Gupta et al., 2024), legal document processing (Narendra et al., 2024), and scientific text understanding. As such, LLMs are increasingly deployed in scenarios requiring comprehension and reasoning over extended, complex documents.

However, current LLMs exhibit fundamental limitations when applied to long-form inputs. A primary bottleneck lies in the quadratic computational complexity of the self-attention mechanism, which makes full-context modeling prohibitively expensive as input length grows. In addition, most pretrained LLMs are constrained by a relatively short effective context window, beyond which their ability to capture long-range dependencies degrades sharply. These limitations are especially detrimental in tasks that demand holistic understanding of long sequences, such as multi-section summarization or cross-paragraph retrieval, where truncation or naive segmentation strategies can lead to loss of critical information and degraded performance.

To address these challenges, various strategies have been proposed for extending the context length of LLMs. One line of work focuses on modifying positional encoding schemes and training dynamics to improve extrapolation: YaRN (Peng et al., 2024) interpolates Rotary Position Embeddings to generalize beyond training lengths; PoSE (Zhu et al., 2024) employs skip-wise positional training to simulate longer contexts using shorter windows; and LongLoRA (Chen et al., 2024) adopts parameter-efficient fine-tuning with sparse attention patterns. Another line of work emphasizes inference-time efficiency: StreamingLLMs (Xiao et al., 2024) cache and reuse representations of earlier tokens, maintaining constant memory usage during streaming generation. Despite these ad-

vances, key limitations persist. YaRN, PoSE, and LongLoRA rely on dense global attention during inference, maintaining quadratic cost with respect to sequence length. StreamingLLMs, while efficient, primarily rely on static token retention strategies that favor early-context anchoring, limiting their ability to capture information that appears in the middle or end of a long input. Moreover, none of these methods incorporate mechanisms to dynamically determine which parts of the input are most relevant to the current prediction—a crucial capability for modeling semantically heterogeneous and structurally diverse documents.

We argue that effective long-context understanding requires not only scalable computation, but also *adaptive information routing* and *multi-granular semantic representation*. Motivated by the human reading process—where readers first build a structural overview before attending to relevant details—we propose to embed a similar hierarchical abstraction and refinement capability within the model architecture. Specifically, we posit that introducing learnable summary representations and structured routing mechanisms can enable LLMs to selectively focus on informative content, while maintaining overall efficiency.

We propose STAR (*Summary Token-guided Attention and Routing*), a novel hierarchical attention architecture that effectively balances resource requirements and expressiveness in long-document modeling. At its core, STAR introduces a learnable summary token `[s]` appended to each chunk of the input sequence. This token serves a dual role: capturing chunk-level semantic information and functioning as a routing signal to guide long-range dependency modeling. The architecture comprises three synergistic stages: (1) Intra-Chunk Abstraction, where self-attention is confined within fixed-length chunks to compute local representations and summarize chunk-level semantics via a learnable summary token `[s]`; (2) Inter-Chunk Routing, where each token performs coarse-grained attention over the summary tokens of preceding chunks to identify semantically relevant regions; (3) Token-Level Refinement, where fine-grained attention is applied over a small subset of retrieved chunks to incorporate detailed information.

The outputs from these three stages are fused via a learnable gating mechanism that adaptively balances local detail, global abstraction, and retrieved precision. This design enables STAR to scale to long sequences with linear or sub-quadratic cost, while preserving the model's ability to reason over semantically important content across distant contexts.

Our main contributions are as follows:

- We propose STAR, a hierarchical attention architecture for long-context modeling that integrates chunk-level abstraction, summary-guided routing, and selective token-level refinement, scaling to long contexts under constrained computational budgets while preserving expressive capacity.
- We introduce a novel use of learnable summary tokens that serve dual roles as semantic aggregators and dynamic routing anchors, enabling relevance-aware cross-chunk retrieval under causal attention constraints.
- We conduct extensive experiments on multiple long-document benchmarks, demonstrating that STAR achieves state-of-the-art performance while significantly reducing computational overhead compared to existing approaches.

## 2 RELATED WORK

The challenge of extending context windows in LLMs has garnered significant attention in recent years, with various approaches emerging to address the computational and memory constraints inherent in processing longer sequences. This section reviews the key methodologies that have been proposed to tackle this fundamental limitation.

### 2.1 POSITION ENCODING MODIFICATIONS

YaRN (Yet another RoPE extensioN method) (Peng et al., 2024) represents a compute-efficient approach to extending context windows of transformer-based language models that utilize Rotary Position Embeddings (RoPE) (Su et al., 2024). The method addresses the fundamental limitation that models fail to generalize past the sequence length they were trained on, requiring 10x fewer tokens

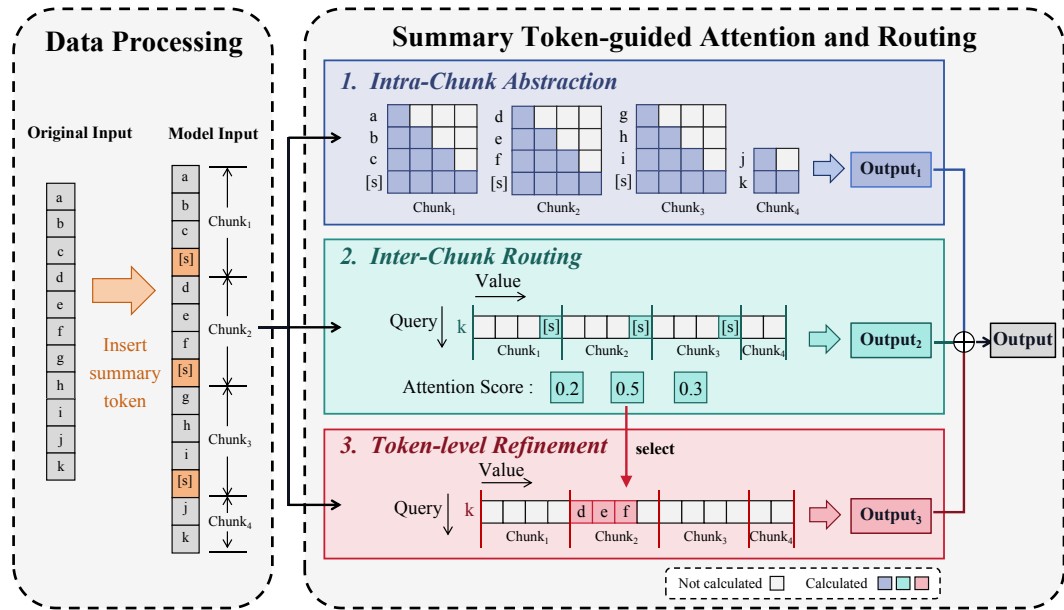

Figure 1: The overall architecture of the STAR (Summary Token-Guided Attention and Routing) method, illustrated using token $k$ (located in the fourth chunk) as an example. The input sequence is first divided into fixed-length chunks, each augmented with a learnable summary token `[s]` appended to its end to represent chunk-level semantics. These augmented chunks serve as the model input. The STAR attention mechanism consists of three stages: (1) Intra-Chunk Abstraction: Self-attention is applied locally within each chunk. This stage generates a local representation for token $k$ ($Output_1$) and enables the `[s]` token to encode the aggregated semantics of its corresponding chunk. (2) Inter-Chunk Routing: Token $k$ (as a query) attends to the summary tokens of all causally preceding chunks (used as keys and values). This stage produces a coarse-grained representation for token $k$ ($Output_2$) and identifies the most relevant chunk(chunk 2 in this example) based on the attention scores, which act as routing signal. (3) Token-Level Refinement: Guided by the routing signal, token $k$ performs an attention computation over the original tokens in the selected relevant chunk (e.g., "$d, e, f$" from $chunk_2$), resulting in a fine-grained representation for token $k$ ($Output_3$). Finally, the three representations are weighted and fused to obtain the final contextual representation of token $k$.

and 2.5x fewer training steps than previous methods. YaRN achieves this efficiency by modifying the RoPE mechanism to better handle longer sequences without requiring full-length fine-tuning on the target context window size.

PoSE (Positional Skip-wisE training) (Zhu et al., 2024) introduces an innovative training strategy that decouples the training sequence length from the target context window size. The method smartly simulates long inputs using a fixed context window by dividing the original context window into several chunks and designing specific position manipulation techniques. PoSE manipulates position indices during training, enabling efficient adaptation to extremely long context windows. This approach significantly reduces the computational overhead associated with full-length fine-tuning while maintaining model performance on extended contexts.

## 2.2 STREAMING AND ATTENTION-BASED APPROACHES

Attention Sinks focuses on efficient streaming language model inference by identifying and preserving critical attention patterns that maintain model coherence across long sequences (Xiao et al., 2024). This method recognizes that certain tokens, particularly initial tokens, serve as "attention sinks" that accumulate attention weights and are crucial for maintaining the model's ability to process subsequent tokens effectively.

LM-Infinite (Han et al., 2024) addresses the challenge of zero-shot extreme length generalization, enabling language models to handle sequences far beyond their training length without additional fine-tuning. This approach leverages architectural modifications and attention mechanisms to maintain performance consistency across varying sequence lengths.

## 2.3 PARAMETER-EFFICIENT FINE-TUNING APPROACHES

LongLoRA (Chen et al., 2024) tackles the challenges of long-context adaptation via a parameter-efficient framework that significantly reduces resource demands compared to full fine-tuning. Building on LoRA, it introduces a shifted sparse attention mechanism that lowers computational complexity while preserving the ability to model long-range dependencies. By systematically shifting sparse attention patterns across layers, the model achieves full sequence coverage with reduced per-layer cost. LongLoRA also incorporates extended-context positional embeddings to ensure stable adaptation to longer inputs. These innovations enable efficient context window extension with minimal overhead, making the method well-suited for resource-constrained settings. Experiments demonstrate that it matches or even surpasses full fine-tuning performance in long-context scenarios, highlighting the potential of parameter-efficient techniques in scalable LLM adaptation.

## 3 STAR

To address the dual challenges of high computational cost and degraded comprehension in long-document processing by LLMs, we propose **STAR** (*Summary Token-Guided Attention and Routing*), a hierarchical attention architecture designed to efficiently handle extended contexts.

STAR is inspired by the cognitive process of human reading: one first abstracts local meaning, then forms a high-level overview, and finally focuses on critical details. Similarly, STAR decomposes long-sequence attention into three stages: (1) **Intra-Chunk Abstraction**, which encodes local semantics within each chunk; (2) **Inter-Chunk Routing**, which identifies semantically relevant regions through summary token interactions; and (3) **Token-Level Refinement**, which applies focused attention to salient content.

### 3.1 CHUNKING AND SUMMARY TOKEN INSERTION

Encoder-based models such as BERT (Devlin et al., 2019) have demonstrated the effectiveness of using a special token (e.g., `[CLS]`) to aggregate sentence-level semantics via bidirectional attention, enabling a single vector to represent an entire sequence. In contrast, autoregressive decoder models, constrained by causal attention, generally do not employ such a mechanism. Drawing inspiration from the encoder paradigm, we argue that representing text chunks with compact vectors offers a powerful strategy to alleviate the challenges posed by long documents.

Our method introduces a learnable vector, the summary token `[s]`, which acts as a compact representation of the semantics for each chunk. The preprocessing pipeline can be formally described as follows:

First, let the input document $D$ be represented as a sequence of $L$ tokens:

$$D = (t_1, t_2, \ldots, t_L) \tag{1}$$

where $t_j$ represents the $j$-th token in the document.

Next, we partition $D$ into $N$ non-overlapping chunks, each of fixed length $k$, and append a summary token `[s]` to the end of each chunk. This transforms the original document into a sequence of augmented chunks, denoted as $D'$:

$$D' = (c'_1, c'_2, \ldots, c'_N) \tag{2}$$

where $N = [L/k]$ is the total number of chunks. Finally, each individual chunk $c'_i$ in the sequence $D'$ consists of $k$ tokens from the original document, followed by the summary token `[s]`:

$$c'_i = (t_{(i-1)k+1}, t_{(i-1)k+2}, \ldots, t_{ik}, [s]) \tag{3}$$

The purpose of this design is to construct STAR's three-level attention representation for each processed token by dynamically fusing local, coarse-grained, and fine-grained contexts. The process unfolds as follows:

- **Intra-Chunk Abstraction (Local Representation)**: The first stage (Sec. 3.2) computes local attention within each chunk to generate local representations. This process allows the complete semantics of all tokens within the chunk to be fused into its trailing `[s]` summary token.

- **Inter-Chunk Routing (Coarse-Grained Representation)**: In the second stage (Sec.3.3), each token attends to the `[s]` summary tokens of the preceding chunks, producing a coarse-grained representation of the long-range context. The attention scores computed in this step are used to identify the most relevant preceding chunks for the current token.

- **Token-level Refinement (Fine-Grained Representation)**: In the final stage (Sec.3.4), guided by the relevance scores from the routing stage, the current token performs a detailed attention pass over the full content of only the top-$k$ most relevant retrieved chunks. This yields a fine-grained representation that concentrates computational resources on the most salient information.

The final output representation for each token is obtained through a weighted fusion of the three distinct attention levels, enabling the model to capture a comprehensive, multi-granular understanding of the context.

## 3.2 INTRA-CHUNK ABSTRACTION

The Intra-Chunk Abstraction stage is designed to efficiently compute local representations by restricting the self-attention mechanism to operate independently within each chunk. This chunk-wise attention avoids the quadratic computational cost of global self-attention, making the approach scalable for long documents.

For each augmented chunk $c'_j$, we perform a local self-attention calculation. Let the input embeddings for the tokens in chunk $c'_j$ be represented by the matrix $E_i \in \mathbb{R}^{(k+1) \times d_{\text{model}}}$. The local self-attention operation computes the chunk's internal hidden states, $h_{\text{local},j}$, as follows:

$$h_{\text{local},j} = \text{Attention}(E_j) \tag{4}$$

This computation produces updated token embeddings within the chunk. In particular:

- Each token in the document obtains a local representation conditioned on the intra-chunk context.
- The summary token `[s]` absorbs the semantics of the entire chunk, yielding a summary vector $s_j$ representing that chunk.

These chunk-wise summary vectors are cached and later used as the coarse-level routing targets in the next stage.

## 3.3 INTER-CHUNK ROUTING

To efficiently incorporate long-range dependencies, STAR introduces an inter-chunk attention mechanism that allows each token to selectively attend to summary tokens of preceding chunks. Let token $t_j$ in chunk $c'_i$ have a query vector $q_j$ derived from its input hidden state $x_j$. Define the set of routing candidates as the summary tokens of earlier chunks:

$$S_{\text{prefix}} = (s_1, s_2, \ldots, s_{i-1})$$

We construct the key-value pairs $(K_{\text{coarse}}, V_{\text{coarse}})$ from $S_{\text{prefix}}$, and compute:

$$h_{\text{coarse},j}, \alpha_j = \text{Attention}(q_j, K_{\text{coarse}}, V_{\text{coarse}}) \tag{5}$$

Here, $h_{\text{coarse},j}$ encodes coarse-grained contextual information gathered from relevant summary tokens, while the attention weights $\alpha_j$ quantify the relative importance of prior chunks. These weights serve a second role: routing signals for the fine-grained refinement step.

## 3.4 TOKEN-LEVEL REFINEMENT

The Token-level Refinement stage computes a fine-grained representation by applying focused attention over the most relevant chunks from earlier chunks in the document. This stage is guided by the routing attention weights $\alpha_j$, which are produced by the Inter-Chunk Routing mechanism and indicate the relevance of preceding chunks to the current token.

First, we identify the indices, $I_{topK}$, of the top-K chunks corresponding to the highest attention scores in $\alpha_j$. The keys ($K_{fine}$) and values ($V_{fine}$) for this refinement stage are then constructed exclusively from the detailed, token-level hidden states of these top-K selected chunks. To do this, we retrieve the hidden state matrices $\{c_m' \mid m \in I_{topK}\}$ and use their contents to form the keys $K_{fine}$ and values $V_{fine}$.

Using the same shared query vector $q_j$ as the other attention mechanisms in the layer, the fine-grained representation, $h_{fine,j}$, is then computed:

$$h_{fine,j} = \text{Attention}(q_j, K_{fine}, V_{fine}) \tag{6}$$

By selectively attending to the full content of only the most relevant chunks, this mechanism injects specific, detailed information into the token's representation. This complements the high-level context provided by the coarse-grained attention, enabling a more nuanced and accurate understanding.

## 3.5 HIERARCHICAL ATTENTION FUSION

The final step fuses the three representations—local, coarse, and fine—into a unified contextual embedding. To control this fusion, we introduce three trainable scalar weights $\{w_{local}, w_{coarse}, w_{fine}\}$ shared across all tokens. These are normalized via softmax to form a set of gates:

$$(g_{local}, g_{coarse}, g_{fine}) = \text{Softmax}(w_{local}, w_{coarse}, w_{fine}) \tag{7}$$

The final output representation for token $t_j$, denoted $h_{final,j}$, is then computed as the weighted sum of the three attention outputs:

$$h_{final,j} = g_{local} \cdot h_{local,j} + g_{coarse} \cdot h_{coarse,j} + g_{fine} \cdot h_{fine,j} \tag{8}$$

This fusion mechanism enables the model to adaptively weigh different levels of abstraction, integrating local context, global structure, and targeted detail. The resulting vector $h_{final,j}$ is then passed to subsequent Transformer blocks for downstream modeling.

Table 1: Perplexity of TinyLLaMA-1.1B under various training methods, evaluated on the GovReport and ProofPile datasets with context window sizes ranging from 2K to 64K tokens. Our STAR approach is trained with a fixed target length of 32K tokens. At inference time, it retrieves eight relevant blocks (each consisting of 512 tokens) as contextual input, enabling effective scaling to 64K-token contexts under limited computational resources.

| Methods | Reference | Gov-report | | | | | | Proof-pile | | | | | |
|---|---|---|---|---|---|---|---|---|---|---|---|---|---|
| | | 2k | 4k | 8k | 16k | 32k | 64k | 2k | 4k | 8k | 16k | 32k | 64k |
| Original | - | **6.43** | 131.37 | $>10^3$ | $>10^3$ | $>10^3$ | $>10^3$ | **5.72** | 135 | $>10^3$ | $>10^3$ | $>10^3$ | $>10^3$ |
| YaRN | ICLR 2024 | 9.56 | 9.53 | 9.61 | 9.96 | 9.96 | 10.9 | 8.54 | 7.76 | 6.75 | 5.73 | 5.16 | 5.05 |
| LM-Infinite | NAACL 2024 | 9.56 | 9.69 | 9.76 | 10.03 | 9.85 | 9.57 | 8.54 | 8.03 | 7.10 | 6.08 | 5.56 | 5.10 |
| StreamingLLM | ICLR 2024 | 6.71 | 6.64 | 6.63 | 6.72 | 6.61 | **6.51** | 6.17 | 5.77 | 5.15 | 4.43 | 4.11 | 3.84 |
| LongLoRA | ICLR 2024 | 10.59 | 10.00 | 9.58 | 9.51 | 9.22 | 9.07 | 10.84 | 8.94 | 7.40 | 6.05 | 5.31 | 4.80 |
| STAR(ours) | This paper | 6.74 | **6.41** | **6.31** | **6.43** | **6.39** | 6.52 | 6.34 | **5.53** | **4.75** | **4.04** | **3.72** | **3.61** |

## 4 EXPERIMENTS

This section presents experiments to evaluate the effectiveness and efficiency of STAR. We first describe the datasets, baselines, and evaluation setup. We then report results on language modeling performance across various context lengths, followed by an analysis of inference-time memory usage.

Table 2: Perplexity of TinyLLaMA-1.1B under various training methods, evaluated on PG-19 and Books3 with context window sizes ranging from 32K to 96K tokens. STAR adopts the same training and inference configuration as in Table 1.

| Methods | Reference | PG-19 | | | | | Books3 | | | | |
|---|---|---|---|---|---|---|---|---|---|---|---|
| | | 32k | 48k | 64k | 80k | 96k | 32k | 48k | 64k | 80k | 96k |
| Original | - | $>10^3$ | $>10^3$ | $>10^3$ | $>10^3$ | $>10^3$ | $>10^3$ | $>10^3$ | $>10^3$ | $>10^3$ | $>10^3$ |
| YaRN | ICLR 2024 | 28.72 | 29.49 | 30.14 | 30.62 | 34.21 | 24.33 | 25.61 | 26.39 | 26.79 | 29.83 |
| LM-Infinite | NAACL 2024 | 27.40 | 27.60 | 27.68 | 27.69 | 27.69 | 22.21 | 22.75 | 22.92 | 22.81 | 22.58 |
| LongLoRA | ICLR 2024 | 22.87 | 22.87 | 23.08 | 30.97 | 31.30 | 17.18 | 17.46 | 17.71 | 22.83 | 22.79 |
| PoSE | ICLR 2024 | 12.68 | 13.61 | 14.63 | 15.60 | 16.62 | 11.46 | 12.61 | 13.75 | 14.62 | 15.40 |
| STAR(ours) | This paper | **11.44** | **11.63** | **11.87** | **12.12** | **12.36** | **10.24** | **10.57** | **10.83** | **10.98** | **11.10** |

## 4.1 EXPERIMENTAL SETUP

We evaluate STAR under a standardized setup, including datasets, baselines, implementation details, and metrics, to enable fair comparison in both modeling quality and efficiency.

**Datasets.** To comprehensively evaluate the effectiveness of STAR, we use several widely adopted language modeling benchmarks, each designed to test different aspects of long-context understanding:

- **GovReport (Huang et al., 2021):** A collection of long-form government reports featuring formal structure and dense dependencies across sections. It tests the model's ability to preserve coherence and capture long-range information in structured documents.

- **Proof-pile (Azerbayev et al., 2022):** A corpus of academic papers and scientific literature. This dataset challenges models with complex reasoning, technical vocabulary, and the need to integrate information across multiple sections.

- **PG-19 (Rae et al., 2020):** A dataset of classic books from Project Gutenberg. Its stylistic diversity and narrative complexity make it well-suited for evaluating coherence and long-range dependency modeling in literary texts.

**SOTA.** We compare our proposed STAR architecture against several state-of-the-art methods for long-context language modeling, including YaRN, PoSE, StreamingLLM, LM-Infinite, and LongLoRA. The following summarizes the key ideas and mechanisms behind each baseline:

- *YaRN* (Peng et al., 2024) extends RoPE-based models with a ramp function combining linear and NTK interpolations (Peng & Quesnelle, 2023), and uses temperature scaling to reduce distribution shifts, allowing effective extrapolation beyond pretrained lengths.

- *PoSE* (Zhu et al., 2024) splits the context into chunks and adjusts position indices via a dynamic jump bias. This reduces fine-tuning cost while extending the context window up to 64k tokens with little performance loss.

- *StreamingLLM* (Xiao et al., 2024) enables training-free long-sequence inference by retaining key-value states as "attention sinks" and using a rolling cache, supporting scalable processing of millions of tokens with low overhead.

- *LongLoRA* (Chen et al., 2024) improves LoRA with Shifted Sparse Attention during training and full attention at inference. Trainable normalization and embeddings enhance adaptation to long contexts with modest compute and memory.

- *LM-Infinite* (Han et al., 2024) introduces a $\Lambda$-shaped mask and distance ceiling so tokens attend to both early and recent context within bounds, preventing logit explosion and enabling length generalization without fine-tuning.

**Implementation Details.** In the experiments, we trained TinyLLaMA-1.1B-chat (Zhang et al., 2024a) on a next-token prediction task for one epoch, comprising 2,600 steps with a global batch

size of 8. The learning rate was set to $2e^{-5}$ with a linear scheduling strategy, including 10 warmup steps. We utilized the Adam optimizer with its default hyperparameters. The fine-tuning dataset was drawn from The Pile (Gao et al., 2020). The training was conducted on a server with Intel(R) Xeon(R) Gold 6258R CPU@2.70GHz and 503G memory, and two NVIDIA A100 Tensor Core GPUs with 80G memory. For evaluation, a single NVIDIA A100 GPU was used, leveraging Flash Attention V2 to facilitate the processing of long documents up to 96k tokens (k=1,024).

Training was performed on a server with an Intel(R) Xeon(R) Gold 6258R CPU @ 2.70GHz, 503 GB RAM, and two NVIDIA A100 GPUs (80 GB each). For inference, we used a single A100 GPU with FlashAttention V2 enabled, allowing efficient processing of sequences up to 96K tokens ($k = 1024$).

**Evaluation Metrics.** We evaluate our model using two key metrics: perplexity and GPU memory usage.

- **Perplexity** is used to measure the language modeling quality. It quantifies the model's ability to predict the next token in a sequence, and is formally defined as the exponential of the average negative log-likelihood over the evaluation set:

$$\text{Perplexity} = \exp\left(-\frac{1}{N}\sum_{i=1}^{N} \log P(t_i \mid t_{<i})\right) \tag{9}$$

  where $t_i$ denotes the $i$-th token, $t_{<i}$ is the preceding context, and $N$ is the total number of tokens in the evaluation sequence. Lower perplexity indicates stronger predictive capability and better understanding of long-range dependencies. We adopt perplexity as it is a standard and widely accepted metric in language modeling, allowing for fair and interpretable comparisons with prior work.

- **Memory Usage** is measured as the peak GPU memory consumption during inference. This metric reflects the computational efficiency and scalability of each method, particularly important when dealing with long-context inputs where standard attention mechanisms incur significant memory overhead.

Together, these metrics provide a balanced evaluation of both modeling effectiveness and resource efficiency.

## 4.2 EFFECTIVENESS OF LANGUAGE MODELING

To evaluate the effectiveness of STAR in long-context language modeling, we compare it against a set of strong baseline methods, including YaRN, LM-Infinite, StreamingLLM, and LongLoRA, across four benchmark datasets: GovReport, ProofPile, PG-19, and Books3. We assess performance using perplexity under varying context window sizes, ranging from 2K to 96K tokens.

Table 1 presents results on GovReport and ProofPile, with context lengths from 2K to 64K tokens. The sliding window technique, as introduced by Press et al. (Press et al., 2022), was applied with a stride of 1,024 tokens to optimize evaluation efficiency. Notably, while baseline methods such as YaRN and LongLoRA show relatively stable performance up to moderate lengths (16K–32K), their perplexity either saturates or degrades beyond that point. In contrast, STAR consistently achieves the lowest perplexity across all context lengths, demonstrating both strong short-context modeling (e.g., 6.41 on GovReport with 4K tokens) and remarkable robustness as the context grows longer (e.g., 6.52 with 64K tokens). Compared to StreamingLLM, which performs well in very long contexts but lacks intermediate refinement, STAR shows better or comparable performance even at maximum length, highlighting the effectiveness of its hierarchical retrieval and fusion design.

In Table 2, we extend the evaluation to PG-19 and Books3, which require even longer context modeling—up to 96K tokens. Most baseline methods exhibit substantial degradation in this setting, either showing a sharp increase in perplexity or reaching performance plateaus. In contrast, STAR maintains consistently strong performance across all context lengths, achieving the lowest perplexity at every scale—for example, 11.44 on PG-19 and 10.24 on Books3 with 32K-token contexts, and continuing to scale smoothly to 12.36 and 11.10 at 96K tokens, respectively.

These results collectively highlight the effectiveness of STAR in long-context language modeling. By combining coarse-to-fine routing and selective refinement, STAR consistently achieves lower

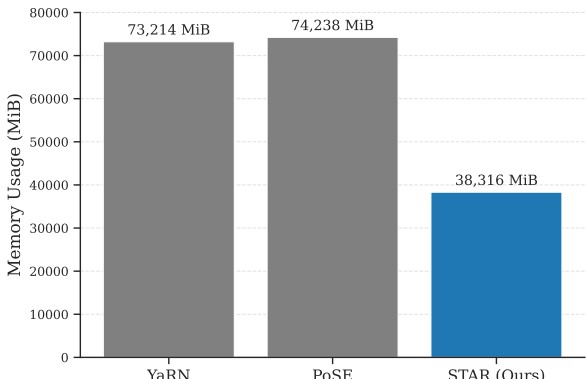

Figure 2: Peak GPU memory usage at 96K-token inference with TinyLLaMA-1.1B. STAR significantly reduces memory consumption compared to other long-context SOTA methods.

perplexity across a wide range of sequence lengths and datasets. This demonstrates its strong ability to capture and integrate semantic dependencies over extended contexts, making it a robust solution for long-document understanding.

### 4.3 MEMORY USAGE

To assess the practical scalability of STAR, we evaluate the inference-time memory efficiency compared to existing long-context methods. Figure 2 presents the peak GPU memory consumption when processing 96K-token sequences using the TinyLLaMA-1.1B model.

The results demonstrate STAR's significant memory efficiency advantages. While state-of-the-art methods such as YaRN and PoSE require substantial GPU memory—73,214 MiB and 74,238 MiB respectively—STAR achieves comparable performance with only 38,316 MiB, representing approximately 48% memory reduction compared to YaRN. This substantial improvement stems from STAR's hierarchical routing mechanism, which circumvents the need for full-sequence global attention by selectively focusing computational resources on the most relevant chunks.

The memory efficiency gains are particularly critical for practical deployment scenarios where GPU resources are constrained. By maintaining superior modeling performance while significantly reducing memory overhead, STAR enables scalable inference on long sequences across a broader range of hardware configurations, making long-context language modeling more accessible for real-world applications.

### 5 CONCLUSION

We propose STAR, a hierarchical attention mechanism that addresses the quadratic complexity bottleneck in long-context language modeling. By introducing learnable summary tokens and a three-stage coarse-to-fine attention process, STAR achieves efficient information routing while preserving the ability to capture long-range dependencies. Our experimental results demonstrate that STAR consistently outperforms existing methods across multiple long-document benchmarks, achieving state-of-the-art perplexity scores on sequences up to 96K tokens while reducing GPU memory usage by approximately 48%. The core insight that effective long-context understanding requires dynamic information routing rather than uniform attention opens new directions for scalable transformer architectures. STAR represents a significant advancement toward practical long-context language modeling, enabling LLMs to process extensive documents efficiently while maintaining high-quality representations essential for comprehensive document understanding.

ETHICS STATEMENT

This work complies with the ICLR Code of Ethics. Our study is methodological in nature and introduces STAR, a hierarchical attention mechanism for efficient long-context language modeling.

All experiments are conducted on publicly available datasets, with no involvement of human subjects, personally identifiable information, or sensitive/private data.

While STAR improves efficiency and scalability, it does not directly mitigate risks of bias or harmful content that may be present in large-scale pretraining corpora. We acknowledge these risks as ongoing challenges and emphasize the importance of responsible deployment.

Finally, by lowering computational cost, STAR may help reduce the environmental impact of training and inference, though its broader societal implications depend on downstream applications. This work was carried out in accordance with principles of research integrity, with no conflicts of interest or external sponsorship.

REPRODUCIBILITY STATEMENT

We have taken multiple steps to ensure the reproducibility of our results. The STAR architecture and its components are described in detail in Section 3, including the hierarchical attention mechanism and chunk-based routing strategy. Experimental settings, including training configurations, hyperparameters, and evaluation protocols, are documented in Section 4 and further elaborated in the Appendix. All datasets used (e.g., GovReport, ProofPile) are publicly available

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

## A  USE OF LLMS

We used LLMs to assist with polishing the writing of this paper. Their usage was limited to language refinement (e.g., improving clarity, grammar, and readability). All conceptual contributions, technical ideas, designs, and experimental results are entirely the work of the authors.

