# OpenReview forum: "STAR: Summary Token-guided Attention and Routing for Efficient Long-Context Reasoning"
_ICLR.cc/2026/Conference — Submitted to ICLR 2026_

### Official Review · Reviewer_2KZn · 2025-10-19

**Soundness:** 2
**Presentation:** 3
**Contribution:** 3
**Rating:** 2
**Confidence:** 5

**Summary:**

The paper introduces STAR, a hierarchical attention mechanism designed to make LLMs more efficient and effective at processing long contexts. STAR reduces the quadratic cost of standard self-attention by: 1. dividing long text into fixed-length chunks and appending summary tokens to each chunk. 2. applying local attention within each chunk to capture chunk-level semantics and embed them in the summary token. 3. each token attends to the summary tokens of preceding chunks to identify relevant chunks for the current token. 4. applying final attention pass on top-k most relevant retrieved chunks. 5. computing final output representation as the weighted sum
of the three attention outputs.
Authors claim that the proposed hierarchical design achieves sub-quadratic scaling in sequence length and greatly reduces memory consumption, thus enabling effective reasoning over long contexts.
Authors benchmark 4 large popular training datasets - GovReport, ProofPile, PG-19, Books3 - in a continuous learning setup to show that STAR achieves the lowest perplexity across all tested context lengths (2K–96K tokens) on TinyLLaMA-1.1B model, surpassing methods such as YaRN, PoSE, StreamingLLM, LongLoRA, and LM-Infinite.

**Strengths:**

1. Authors study the important problem in modern LLMs - efficient long-context training.
2. The STAR framework introduces a novel three-stage hierarchical attention process.
3. Authors show up to ~40% lower GPU memory usage compared to leading methods (e.g., YaRN and PoSE).
4. Authors show perplexity improvements across several diverse training corpora within a wide rage of context lengths.

**Weaknesses:**

1. While the overall method is novel, there are a large number of works that utilize chunk-level attention and/or recombination of local and global attention to improve long-context understanding [1-6]. The paper is missing to provide references for any such works, i.e. not necessary references provided below, but any works in this area which are plenty.
2. The choice of chunk size (k) and number of relevant chunks top-k should directly affect the performance, but authors don't provide any ablation study on these or other hyperparameters introduced in this method.
2a. Index k used multiple times across the paper with different meaning, which is confusing: in Fig 2 ("using token k"); chunk length (Sec 3.1 and further); and in top-k (sec 3.4 and further)
3. The evaluation focuses mainly on *in-distribution* language modeling perplexity over text datasets and lacks downstream reasoning evaluation, as well as out-of-distribution evaluation.
4. Generalizability to other architectures and scales remains unexplored.
5. While more efficient in runtime, STAR introduces a multi-stage computation pipeline. The paper is missing comparison in training and inference costs.
6. A deeper analytical justification could strengthen the conceptual claims about scalability and expressiveness
7. Because STAR only refines over a subset of selected chunks, and on the inter-chunk attention estimation quality, there is room for global consistency loss and information loss during weighting and selection process, especially in highly interdependent or non-linear texts (e.g., dialogues, code). Thus in my opinion further investigation of theoretical bounds, hyperparameter setup, and downstream performance is crucial in this line of works.


References:
1. An, Chenxin, et al. "Training-free long-context scaling of large language models." arXiv preprint arXiv:2402.17463 (2024)
2. Xie, Jiawen, et al. "Chunk, align, select: A simple long-sequence processing method for transformers." arXiv preprint arXiv:2308.13191 (2023).
3. Zhu, Zhenhai, and Radu Soricut. "H-transformer-1d: Fast one-dimensional hierarchical attention for sequences." arXiv preprint arXiv:2107.11906 (2021).
4. Wang, Shuohang, et al. "Cluster-former: Clustering-based sparse transformer for long-range dependency encoding." arXiv preprint arXiv:2009.06097 (2020).
5. Guo, Yuxiang. "SCCA: Shifted Cross Chunk Attention for long contextual semantic expansion." arXiv preprint arXiv:2312.07305 (2023)
6. Xiao, Chaojun, et al. "Infllm: Training-free long-context extrapolation for llms with an efficient context memory." Advances in Neural Information Processing Systems 37 (2024): 119638-119661

**Questions:**

1. You refer to the Appendix for experimental details (line 507), but there was no appendix provided

---

### Official Review · Reviewer_fJcY · 2025-10-30

**Soundness:** 2
**Presentation:** 2
**Contribution:** 1
**Rating:** 2
**Confidence:** 4

**Summary:**

This paper introduces STAR, a hierarchical attention mechanism for long-context LLMs that reports good empirical results on perplexity (Tables 1, 2) and a decent reduction in inference memory (Figure 2). However, the paper's contribution is severely undermined by two major issues. First, there is a critical lack of originality; the proposed three-stage method appears to be a reimplementation of the Landmark Attention (https://arxiv.org/abs/2305.16300) paper, also a line of prior work that is neither cited nor discussed in the related work section (Section 2). This omission makes it impossible to assess any true novelty. Second, the paper suffers from a lack of technical rigor, failing to specify critical implementation details about key architectural components, such as whether the fusion weights (Section 3.5) and the summary learnable vectors (Section 3.1) are shared across layers or heads.

**Strengths:**

The paper's strengths are primarily in its clarity and the quality of its evaluation.

## Visual Clarity
The paper's core idea is presented with a clear and intuitive visualization (Figure 1). This figure effectively illustrates the proposed three-stage architecture (Intra-Chunk Abstraction, Inter-Chunk Routing, and Token-Level Refinement), making the high-level concept easy to grasp.

## Quality
The paper provides a decent empirical evaluation. The authors benchmark their method against a range of baselines. A key strength of the evaluation is the inclusion of not just performance metrics (perplexity) but also practical efficiency metrics, specifically the peak GPU memory usage comparison, which directly supports the paper's claims of scalability.

**Weaknesses:**

## Major Concerns Regarding Novelty and Originality
- The core mechanism of STAR—using special summary/landmark tokens to represent chunks, performing a coarse-grained attention over these tokens, and then a fine-grained attention over the original tokens in selected chunks (Sections 3.1-3.4)—is highly derivative of the "landmark attention" concept which is proposed in Landmark Attention (https://arxiv.org/abs/2305.16300) in 2023 Nov.
- The related work (Section 2) conspicuously omits any mention of this highly relevant line of work on hierarchical and landmark-based sparse attention such as Hierarchical Document Transformer (https://arxiv.org/abs/2407.08330), Hierarchical Sparse Attention (https://arxiv.org/abs/2504.16795) and the Landmark Attention as mentioned before. The current literature review is narrowly focused and fails to situate STAR within the proper context of existing, similar architectures.
- Due to this omission, the paper's central claim of novelty is unsubstantiated. The empirical results are presented in a vacuum without comparison to the most directly related prior art, making the paper's true contribution unclear.

## Missing Technical Details on Fusion Mechanism
- The paper proposes a fusion of three representations ($h_{local,j}$, $h_{coarse,j}$, $h_{fine,j}$) using learnable scalar weights $\{w_{local}, w_{coarse}, w_{fine}\}$ (Section 3.5, Eq. 7 & 8). And summary tokens [s] as input are "learnable vectors" as mentioned in the manuscript. But the manuscript provides no information on how these weights are parameterized. It is unclear if they are shared across all layers, shared across all attention heads within a layer, or if each head in each layer learns an independent set of fusion weights.

- This is a critical architectural detail required for reproducibility and for understanding the model's complexity and parameterization.

**Questions:**

- The core mechanism of STAR appears substantially similar to the 2023 Landmark Attention (https://arxiv.org/abs/2305.16300) paper. Could the authors please clarify the specific, novel technical contributions of STAR that differentiate it from this prior work?
- Regarding the fusion mechanism, what is the parameter-sharing strategy for the weights $\{w_{local}, w_{coarse}, w_{fine}\}$? Are they shared across all layers and heads, or learned per-layer/per-head?
- How is the "learnable vector" for the summary token [s]  implemented? Is this a single, globally shared embedding vector, or is it parameterized differently?
- Can the authors justify the omission of highly relevant methods like Landmark Attention, Hierarchical Document Transformer, and Hierarchical Sparse Attention from the related work section?

---

### Official Review · Reviewer_7WXo · 2025-10-31

**Soundness:** 2
**Presentation:** 2
**Contribution:** 2
**Rating:** 2
**Confidence:** 4

**Summary:**

The paper proposes STAR (Summary Token-guided Attention and Routing), a framework that addresses the challenge of processing long contexts in LLMs caused by the quadratic computational complexity of self-attention. To reduce this complexity, tokens are partitioned into chunks and processed through a three-stage pipeline: Intra-Chunk Abstraction, Inter-Chunk Routing, and Token-Level Refinement. In the Intra-Chunk Abstraction stage, a summary token for each chunk captures local semantics. In the Inter-Chunk Routing stage, queries are routed to the most relevant chunks. In the Token-Level Refinement stage, token-level attention is applied only to the selected chunks to refine contextual representations. As a result, STAR lowers computational cost while preserving long-context performance.

**Strengths:**

- The STAR framework processes inputs by chunking rather than attending to the entire context at once, enabling more efficient handling of all tokens. It handles longer inputs (e.g., PG-19, Books3) than comparable baselines while consistently achieving lower perplexity, demonstrating robust performance across 32K–96K tokens.

- Chunking long-form inputs reduces peak GPU memory consumption by about 48% compared with the baseline.

**Weaknesses:**

- The reported long-context gains are demonstrated only on a 1B-parameter LLM. However, most long-context baselines are typically evaluated at 7B+ LLM scales. Therefore, the method should demonstrate how much compute and memory savings carry over to larger scales (7B+). Without such evidence, it remains uncertain whether the same benefits persist or improve at scale.

- There is no ablation study analyzing how each component (e.g., intra-chunk summarization, inter-chunk routing, token-level refinement, chunk size) contributes to long-context performance, nor how sensitive the gains are to these design choices and compute/memory budgets.

- The study reports only a single-point measurement of peak GPU memory at 96K tokens. Because it remains unclear which context-length ranges deliver the observed advantage, please provide results at 32K, 48K, 64K, 80K, and 96K, and either visualize or quantify where the GPU memory peak occurs.

- The evaluation relies primarily on weak baselines (e.g., PoSE/YaRN) and does not include comparisons with more recent long-context methods [1–3]. To ensure fairness and recency, the assessment should include the latest models.



[1] ICML’25 : LongRoPE2: Near-Lossless LLM Context Window Scaling

[2] ACL’25 : LADM: Long-context Training Data Selection with Attention-based Dependency Measurement for LLMs

[3] ICLR’25 : TidalDecode: Fast and Accurate LLM Decoding with Position Persistent Sparse Attention

**Questions:**

- It would be helpful to include long-context results on 7B+ LLMs using the same methodology.
- When compared under identical settings to recent studies, the magnitude of improvement should be reported quantitatively.
- The study should clarify how ablations on Abstraction, Routing, and Refinement affect performance, and what each stage’s contribution is.
- For 32K–96K under identical conditions, the paper should provide the peak GPU memory usage and the reduction relative to the baseline, and analyze where the efficiency gains occur.

---

### Meta-Review · Area_Chair_zCne · 2025-12-08

**Summary:**

The paper introduces STAR, a hierarchical attention mechanism designed to make LLMs more efficient and effective at processing long contexts. The reviewers raised several concerns on the submission. For instance, one reviewer argues that the reported long-context gains are demonstrated only on a 1B-parameter LLM. However, most long-context baselines are typically evaluated at 7B+ LLM scales. Therefore, the method should demonstrate how much compute and memory savings carry over to larger scales (7B+). Without such evidence, it remains uncertain whether the same benefits persist or improve at scale." Neverthless, the authors did not submit responses to these concerns.

**Reviewer Concerns:**

No rebuttal was provided.

**Reviewer Scores:**

No rebuttal was provided.

---

### Decision · Program_Chairs · 2026-01-26

Reject